# Thermo-Responsive Shape Memory Vanillin-Based Photopolymers for Microtransfer Molding

**DOI:** 10.3390/polym14122460

**Published:** 2022-06-16

**Authors:** Justinas Jaras, Aukse Navaruckiene, Edvinas Skliutas, Jurga Jersovaite, Mangirdas Malinauskas, Jolita Ostrauskaite

**Affiliations:** 1Department of Polymer Chemistry and Technology, Kaunas University of Technology, Radvilenu Rd. 19, LT-50254 Kaunas, Lithuania; justinas.jaras@ktu.edu (J.J.); aukse.navaruckiene@ktu.lt (A.N.); 2Laser Research Center, Faculty of Physics, Vilnius University, Sauletekis Ave. 10, LT-10223 Vilnius, Lithuania; edvinas.skliutas@ff.vu.lt (E.S.); jurga.jersovaite@ff.stud.vu.lt (J.J.); mangirdas.malinauskas@ff.vu.lt (M.M.)

**Keywords:** bio-based polymers, dual curing, shape-memory, photocuring, microtransfer molding

## Abstract

Novel thermo-responsive shape-memory vanillin-based photopolymers have been developed for microtransfer molding. Different mixtures of vanillin dimethacrylate with tridecyl methacrylate and 1,3-benzenedithiol have been tested as photocurable resins. The combination of the different reaction mechanisms, thiol-acrylate photopolymerization, and acrylate homopolymerization, that were tuned by changing the ratio of monomers, resulted in a wide range of the thermal and mechanical properties of the photopolymers obtained. All polymers demonstrated great shape-memory properties and were able to return to their primary shape after the temperature programming and maintain their temporary shape. The selected compositions weretested by the microtransfer molding technique and showed promising results. The developed thermo-responsive shape-memory bio-based photopolymers have great potential for forming microtransfered structures and devices applicable on non-flat surfaces.

## 1. Introduction

Increased oil prices and environmental awareness have prompted the scientific community to seek alternative feedstocks for polymers, which nowadays are mainly derived from fossil fuels [1,2]. The main source of sustainable and environmentally friendly polymers are renewable raw materials, such as plants [3]. Currently, vanillin is one of a few bio-based and aromatic compounds that are industrially available [4]. Its derivatives have been successfully used in various polymerization techniques. For example, vanillin acrylate-based polymers, synthesized by thermal polymerization, possess high glass transition temperature and high mechanical strength [5]. Recently, vanillin-based polymers, synthesized by photopolymerization, demonstrated significant antimicrobial activity and were applied in optical 3D printing [6]. However, there are only a few examples of the usage of vanillin derivatives in dual curing systems [7].

Dual curing is a process that combines two curing reactions that occur simultaneously or sequentially [8]. In this process, the properties of the resulting materials can be easily changed by manipulating the composition of the resins by changing the ratio of the selected monomers [9]. Polymers with unique mechanical and thermal properties can be obtained during the dual curing process as the result of interpenetrating, or semi-interpenetrating, polymer networks [10]. Due to these extraordinary features, dual curing attracts huge attention among scientists as one of the possible ways to create smart materials, such as shape-memory polymers [11].

Shape-memory polymers are a part of intelligent polymeric systems, also known as smart materials [12]. Their unique property of changing their shape in response to a stimulus has significantly broadened their areas of application [13]. Bio-based shape-memory polymers attract huge interest in the scientific community, however, only a few examples of UV photocured shape-memory polymers are available [14]. Recently, novel shape-memory bio-based polymers have been successfully synthesized from castor oil with the introduction of dynamic pyrazole–urea bonds [15]. Bovine serum albumin-based shape-memory bioplastics have been produced and adapted for stereolithography [16]. Cholesterol was successfully used in the synthesis of thermo-responsive shape-memory polymers [17]. The novel poly(cholesteryl methacrylate) coatings were characterized as potential substrates for tissue engineering, showing the importance of shape-memory polymers [18]. In this work, we focused our attention on the development of novel bio-based shape-memory photopolymers suitable for the manufacturing of various structures and devices by nanoimprint lithography. 

Photocuring requires a smaller amount of resources to produce the required part compared to other techniques [19]. It is a simple way to create unique products of complex shapes while also reducing manufacturing costs and carbon dioxide emission [20]. The final product can be successfully manufactured during the molding process and does not need further modification before usage [21]. Moreover, even the most complex parts can be duplicated easily if the original part was damaged during the operation process [22]. Microtransfer molding, or nanoimprint lithography, is a great way to reduce waste as it uses a minimal amount of material needed for the final product [23], and it is a versatile technique for nanofabrication of a wide range of micro and nanostructures and devices [24]. It offers the ability to fabricate nanostructures that exhibit vertically aligned nanoarrays and nanopatterns, and the ability to have precise control over the size and geometry of the nanostructures, and to form nanostructures attached to a bulk support. The main advantage of this technique is its fast, easy, and inexpensive, but accurate, reproduction of various 2D or 3D objects from the nanoscale to the macroscale [25,26].

In this work, different mixtures of bio-based monomer vanillin dimethacrylate with tridecyl methacrylate, derived from natural oil and 1,3-benzenedithiol, have been tested to find an efficient photocurable system for microtransfer molding. Ethyl (2,4,6-trimethylbenzoyl) phenylphosphinate was selected as the photoinitiator, due to its ability to cure the deep layers of the resin and its photobleaching effect [27]. This work is a continuation of our previous studies [7,28]. In the present work, the addition of tridecyl methacrylate to the resin composition has led to a reduction in the use of expensive vanillin monomer and, thus, to a reduction in the cost of the polymers, without sacrificing their properties. The polymers obtained demonstrated a wide range of thermal and mechanical properties as a result of the combination of the different reaction mechanisms, thiol-acrylate photopolymerization and acrylate homopolymerization, which were tuned by changing the ratio of monomers. The increase in tridecyl methacrylate content resulted in less rigid polymers, lower photocuring rate, and lower Young’s modulus values. The reduction of thiol content increased the shrinkage of polymers, the photocuring rate, and the thermal and mechanical characteristics of the resulting polymers. All polymers demonstrated shape-memory properties. The most promising compositions have been tested by the microtransfer molding technique and showed promising results as photoresins for nanoimprint lithography. Microtransfered structures and devices formed from the developed thermo-responsive shape-memory bio-based photopolymers can be applied on non-flat surfaces, convex or concave, such as cylinders, tubes, etc.

## 2. Materials and Methods

### 2.1. Materials

Vanillin dimethacrylate (VDM, Specific Polymers, Castries, France), 1,3-benzenedithiol (BDT, Fluorochem, Glossop, United Kingdom), tridecyl methacrylate (C13-MA, VISIOMER^®^ Terra C13-MA, Evonik, Essen, Deutschland), ethyl(2,4,6 trimethylbenzoyl)phenylphosphinate (TPOL, Fluorochem, Glossop, United Kingdom), (Figure 1) were used as received.

### 2.2. Preparation of Cross-Linked Polymer Specimens

The mixtures containing 1 mol of VDM, 3 mol.% of TPOL, 1, 0.75, 0.5, or 0.25 mol of BDT and 1.5, 3, or 4.5 mol of C13-MA were stirred with a magnetic stirrer at room temperature (25 °C) for 1 min. When homogeneous mixtures were obtained, the resins were poured into a Teflon mold and cured for 5–7 min in the UV irradiation chamber BS-02 (Opsytec Dr. Grobel, Ettlinger, Germany) with an intensity of 30 mW/cm^2^ and a wavelength range of 280–400 nm. The composition of the resins is presented in Table 1.

### 2.3. Characterization Techniques

Fourier transformation infrared (FT-IR) spectroscopy spectra were recorded using a Spectrum BX II FT-IR spectrometer (Perkin Elmer, Llantrisant, UK). Reflection was measured during the test. The range of wavenumbers was (650–4000) cm^−1^.

The Soxhlet extraction was used to determine the yield of the insoluble fraction. 0.2 g polymer samples were extracted with acetone for 24 h. After 24 h, the insoluble fractions were dried under vacuum until no weight changes were observed. The yield of insoluble fraction was calculated as the weight difference before and after extraction and drying.

Thermogravimetrical analysis (TGA) was performed on a TGA 4000 apparatus (Perkin Elmer, Llantrisant, UK). A heating rate of 20 °C/min was chosen in a nitrogen atmosphere (100 mL/min). The temperature range of (10−800) °C was used. Aluminium oxide pans were used.

Dynamical mechanical thermal analysis (DMTA) was performed on an MCR302 rheometer (Anton Paar, Graz, Austria) equipped with the SRF10-SN30777 measuring system. The Peltier-controlled temperature chamber was used. The temperature was increased from −80 °C to 100 °C, with a heating rate of 2.99 °C/min. The normal force was set at −0.1 N during the measurement. In all cases, the torsion mode was used with a frequency of 1 Hz and a torsion strain of 0.1%. The storage modulus (G′), the loss modulus (G″), and the loss factor (tanδ) were recorded as functions of temperature. 

The mechanical properties of the synthesized polymers were determined by the tensile test. The tensile test was performed on a Testometric M500-50CT testing machine (Testometric Co. Ltd., Rochdale, UK) with flat-faced grips at room temperature (21.5 °C). The dimensions of the test specimens were 70 (±0.01) × 10 (±0.01) × 2 (±0.15) mm. The gap between the grips was set to 20 mm and the test was performed at a speed of 5 mm/min until the specimen broke. Young’s modulus, tensile strength, and elongation at break were determined. 

### 2.4. Real-Time Photorheometry

UV/Vis curing tests were performed with resins containing 1 mol of vanillin dimethacrylate, 3 mol.% of TPOL, 1, 0.75, 0.5, or 0.25 mol of BDT and 1.5, 3.0, or 4.5 mol of C13-MA on a MCR302 rheometer (Anton Paar, Graz, Austria) equipped with the plate/plate measuring system. The measuring gap was set to 0.1 mm and the samples were irradiated by UV/Vis light in a wavelength range of 250–450 nm through the glass plate using the OmniCure S2000 UV/Vis spot curing system (Lumen Dynamics Group Inc., Mississauga, ON, Canada). The temperature was 25 °C. The shear mode with a frequency of 10 Hz and a shear strain of 1% was used in all cases. The storage modulus (G′), the loss modulus (G″) and the complex viscosity (η*) were recorded as a function of the irradiation time and the values of each parameter taken after 300 s of photocuring are presented in Table 2. The gel point (t_gel_) was calculated as the intersection point of the G′ and G″ curves. The shrinkage was calculated from the reduction of the height of the sample during the polymerization process. The normal force was set at 0 N during the measurement of the sample shrinkage. Five measurements of each resin were used to obtain the mean value and standard deviation. 

### 2.5. Microtransfer Molding Technique

A microtransfer molding (μTM) technique, or nanoimprint lithography, was used to test **C1** and **C8** resins to make replicas [29]. First, a master structure was manufactured out of PlasGray material with the Asiga Pico2 39 UV table-top 3D printer (Asiga, Alexandria, Australia). Next, polydimethylsiloxane (PDMS) was poured over this structure and thermally cured at 100 C for 1 h, thus creating a soft mold (stamp). It was then used to make a replica of both resins. The printed structure was a 1951 USAF resolution target (25% of the downloaded vector file size), used as a sample in optical engineering for testing imaging systems. A UV diode emitting 365 nm wavelength light (CS2010, Thorlabs, Newton, NJ, USA) was used to cure the **C1** and **C8** resins and obtain the replicas. The curing time was set to 5 min for each sample. The soft mold and the replicas made were characterized using the Olympus IX73 optical microscope (Olympus Corporation, Tokyo, Japan).

## 3. Results

### 3.1. Monitoring of Photocuring Kinetics by Real-Time Photorheometry

The photocuring of the three-component resins was studied by real-time photorheometry. In this study, the most important parameters, the photocuring rate (characterized by t_gel_, induction period and slope of the curves), rigidity (characterized by storage modulus G′), and shrinkage (described by volume loss during photopolymerization) have been monitored and examined [30]. The real-time photorheometry data of all resins are summarized in Table 2.

The dependence of the storage modulus G′ on the irradiation time of all resins is presented in Figure 2. The storage modulus represents the stiffness and rigidity of the polymers that are the important parameters for the applications of the polymers. It was determined that the resin composition had a huge influence on the photocuring kinetics and on the properties of the resulting polymers. The most rigid polymer was **C4** prepared with the lowest amount of C13-MA and BDT. The highest amount of VDM in **C4** compared to other polymers caused the high rate of acrylate homopolymerization. After comparing polymer **C4** with **C8** and **C12**, it was determined that the rigidity of the polymers decreased with increasing volume of tridecyl methacrylate in the resins. The rigidity of the polymers was reduced from 17.343 (**C4**) to 2.930 (**C12**) MPa by increasing the content of C13-MA from 1.5 to 4.5 mol. The long carbon chain of tridecyl methacrylate was the reason for the formation of the soft and flexible polymer [31]. By increasing the amount of C13-MA, the amount of polymer to plasticize long flexible alkyl chains was increased, and, thus, the rigidity of the polymer was reduced, while the increase in the amount of VDM, having a relatively short and partially aromatic structure in comparison to C13-MA, resulted in higher rigidity of the polymers.

The amount of thiol also had a great impact on the polymer structure and, therefore, its characteristics. It was determined that the polymer rigidity was increased by reducing the amount of thiol in the composition. The high amount of thiol resulted in a higher rate of thiol-ene photopolymerization, which is a slower reaction in comparison to free radical polymerization. A high number of short polymer chains was formed as a result of competing free radical and thiol-ene photopolymerizations, and even increased with increasing amounts of thiol. However, according to the literature, thiols not only reduce the rigidity of polymers but also increase their flexibility due to the formation of flexible thioether linkages [7]. For example, the rigidity of polymers **C5**–**C8**, prepared with 3 mol of C13-MA, was increased from 0.031 (**C5**) to 10.148 (**C8**) MPa by decreasing the amount of thiol from 1 to 0.25 mol. This tendency was also visible in polymers **C1**–**C4** prepared with 1.5 mol of C13-MA, and polymers **C9**–**C12** prepared with 4.5 mol of C13-MA. It is important to note that different areas of application require different materials, and both, rigid and soft, materials are needed. 

The photocuring rate (described by the t_gel_, induction period, and slope of the curve) is the other important parameter for the optimal curing process. The gel point is extremely important in microtransfer molding technology. The lowest values of t_gel_ and the shortest induction period were demonstrated by polymers prepared with the lowest amount of thiol. The curve of storage modulus of these polymers also reached the plateau faster than that of other vanillin-based polymers. As was stated earlier, free radical photopolymerization is a faster process in comparison to thiol-ene photopolymerization. However, not only the reaction mechanism, but also the structure of the monomer affects the gel point. C13-MA has a much longer structure compared to vanillin dimethacrylate and, as a result of that, the increase in C13-MA in compositions slowed the photopolymerization process, while the increase in VDM made it faster [28]. Subsequently, the lowest t_gel_ value was acquired by resin **C4** (t_gel_ = 3.4 s) prepared with 0.25 mol of thiol and 1.5 mol of tridecyl methacrylate. Resin **C12**, which was prepared with the same amount of thiol and a higher amount of C13-MA, had a much higher gel point (t_gel_ = 9.8 s) compared to resin **C4**, which was the result of an increased amount of flexible tridecyl methacrylate chains. 

The shrinkage is also very important for the application of polymers. Low values of shrinkage are needed to form precise and unique structures, as high shrinkage could result in failed molding attempts. Values as low as 0–5% are mandatory in order to obtain high-quality products [32]. The lowest shrinkage values were demonstrated by polymers prepared with the highest amount of thiol. The shrinkage values of the polymers with 1 mol of thiol and different amounts of tridecyl methacrylate, **C1**, **C5**, and **C9**, were as low as 0–2%. However, resins with a lower amount of thiol and a higher amount of acrylate shrunk more. For example, the shrinkage increased from 1 to 10% as a result of the reduction of the amount of thiol from 1 to 0.25 mol in the resins **C9**–**C12**. This can be explained by the dominant reaction mechanism, as it is well known that acrylates have a high shrinkage rate during free radical polymerization [33], while thiol-ene photopolymerization results in a lower shrinkage volume [34]. That is because long-distance connections via weak Van der Waals force are replaced by short, strong covalent bonds between carbon atoms in monomer units during free radical acrylate polymerization, which results in shrinkage of the polymers [33].

After analyzing all of these results, the **C8** resin was selected as the most promising composition for the microtransfer molding technique. This composition demonstrated one of the lowest gel point values and relatively low shrinkage, which makes the process faster and allows the creation of complex and accurate shapes. Furthermore, the induction period of this resin was less than 1 s and the resultant polymer was a stiff material with a high value of G′. All these properties made the **C8** resin a suitable candidate for the microtransfer molding technique. For comparison, the resin **C1** that forms soft but non-brittle polymer was also selected for microtransfer molding.

### 3.2. Characterization of Photocross-Linked Polymer Structure 

The chemical structure of the photocross-linked polymers was confirmed by FT-IR spectroscopy. Signals of C=O, which were present at 1714–1735 cm^−1^, and those of the C=C group, which were present at 1606–1639 cm^−1^ in VDM and C13-MA spectra, were also reduced in polymer spectra. The signals of the S-H group, which were present at 2560 cm^−1^ in the BDT spectra, were not visible in the polymer spectra and the new signal of the C-S group was detected at 1120–1122 cm^−1^ in the polymer sample spectra. These changes in the polymer spectra indicated the formation of cross-linked structures in the polymers. As an example, the FT-IR spectra of VDM, C13-MA, BDT, and polymers **C5**–**C8** are presented in Figure 3.

The Soxhlet extraction was also performed in order to confirm the cross-linked structure of the polymers. The yield of the insoluble fraction of these polymers was in the range of 61–90% (Table 3). The high yield of the insoluble fraction indicated that the dense polymer network was formed during photopolymerization. However, polymers **C1, C5**, and **C9**, which were prepared with 1 mol of thiol, demonstrated a relatively low value of the yield of the insoluble fraction (61–70%). This was probably due to the large amount of soluble linear or branched polymer fragments, which were formed as the result of spatial hindrances as the polymer chains were unable to pass through each other during the photo polymerization process [35]. Acrylate free radical homopolymerization is faster than thiol-ene photopolymerization, and, as a result of that, the spatial hindrances were increased by increasing the amount of thiol.

### 3.3. Thermal Properties of Cross-Linked Polymers 

Dynamical mechanical thermal analysis (DMTA) and thermogravimetrical analysis (TGA) were used to study the thermal properties of the polymers. The results are summarized in Table 3. DMTA was used to determine the glass transition temperature (T_g_) of the polymer samples. The DMTA curves are presented in Figure 4. The glass transition of the polymers was in the range of −10–54 °C. It was determined that the higher values of T_g_ were obtained when the lower volume of thiol was used. For example, T_g_ increased from −10 °C for polymer **C1** to 54 °C for polymer **C4** by reducing the amount of BDT from 1 to 0.25 mol. The low glass transition temperature was the result of softer polymer with a less dense structure and low yield of the insoluble fraction [36]. The amount of C13-MA also had a great influence on T_g_. For example, the increase in the amount of C13-MA from 1.5 mol in the polymer **C2** to 4.5 mol in the polymer **C10** reduced the T_g_ from 10 to −1 °C. This was the result of a long carbon chain of C13-MA. Long chains made the polymer softer and flexible, and decreased the glass transition temperature of the polymers [37].

The thermal decomposition of the polymers occurred in one step (Figure 5). The temperature of 10% weight loss (T_dec.−10%_) was in the range of 308–346 °C. Consequently, to the glass transition temperature, the temperature of 10% weight loss was reduced by increasing the amount of BDT in the polymer. For example, T_dec.−10%_ increased from 308 °C to 332 °C by decreasing the amount of thiol from 1 to 0.25 mol. The results were correlated with the yield of the insoluble fraction of the polymers, as higher decomposition temperatures were shown by the polymers with the higher yield of the insoluble fraction and the denser inner polymer network. Different results were observed when the amount of C13-MA was increased. There was no consistent increase or decrease in T_dec.−10%_ when the amount of tridecyl methacrylate increased. Very similar values of the temperature of 10% weight loss were observed when 1.5 and 4.5 mol of C13-MA were used. However, polymers prepared with 3 mol of C13-MA demonstrated higher values of 10% weight loss. For example, polymers **C4** and **C12**, which were prepared with 1.5 and 4.5 mol of C13-MA, reached the temperature of 332 °C and 331 °C, respectively, while polymer **C8**, prepared with 3 mol of C13-MA, reached the temperature of 344 °C. 

### 3.4. Thermomechanical Properties of Cross-Linked Polymers

Dynamical mechanical thermal analysis (DMTA) was performed to characterize the mechanical properties of the developed polymers. The dependence of the storage modulus G′ on temperature was measured (Figure 6) and the values of G′ before and after the glass transition temperature were compared (Table 3). For the compatible results, the storage modulus values were taken from the plateau of the curve as the plateau indicates that polymer is in the steady state and that no changes in its structure are happening. The chosen temperatures were −80 °C (before T_g_) and 100 °C (after T_g_). After the results were compared, it was determined that the storage modulus of these polymers decreases greatly as the temperature increases to values higher than those of their T_g_. Because of this, the developed polymers are rigid materials below their glass transition temperature and become soft and flexible when the temperature increases above their T_g_. For example, the storage modulus of polymer **C1** was reduced from 761.99 MPa to 3.36 MPa when the temperature increased from −80 °C to 100 °C. These results correlate to the rheological and thermal characteristics, as the same tendencies occur when these polymers are compared. For example, polymers **C4**, **C8**, and **C12**, prepared with different amounts of C13-MA, show that the storage modulus reduces from 2350.37 MPa to 661.37 MPa (at −80 °C) when the amount of C13-MA increases from 1.5 mol to 4.5 mol, which was the result of an increased amount of flexible tridecyl methacrylate chains, as mentioned above. Increased amounts of BDT in the polymer also reduced the rigidity of the polymers. For example, rigidity was reduced from 902.59 MPa (polymer **C8** at −80 °C) to 386.44 MPa (polymer **C5** at −80 °C) by increasing the amount of thiol from 0.25 mol to 1 mol, due to the formation of flexible thioether linkages, as also mentioned earlier.

### 3.5. Shape-Memory Properties of Cross-Linked Polymers

Thermo-responsive shape-memory polymers attract huge interest in the scientific community because of their unique ability to remember their shape. To show shape-memory properties, polymer samples were heated above their glass transition temperature, reformed to the temporary shape, and cooled down to the temperature lower than their T_g_. All polymers were rigid materials and were able to maintain their temporary shape when the temperature was below their T_g_. To return to a permanent shape, polymer samples were heated above their glass transition temperature, and all polymers were able to return to a permanent shape within seconds. The recovery to original shape conditions of these polymers is determined by their glass transition temperature [14].

Polymers were transformed to their temporary shape at 60 °C temperature and then placed in the refrigerator (−20 °C) to maintain their temporary shape. All developed polymers were able to maintain their permanent shape at a temperature below 0 °C, however, only a few of them were able to maintain it at room temperature (25 °C). Because of this, the application of these polymers is limited. The most promising results were demonstrated by the polymers **C3**, **C4,** and **C8**, because these polymers have relatively high glass transition temperature (from 28 to 54 °C) and can maintain their temporary shape at room temperature and above. The transformation of the **C8** sample is presented in Figure 7.

### 3.6. Mechanical Characteristics of Cross-Linked Polymers

The tensile test was performed to investigate the mechanical properties of the obtained polymers. The results are presented in Table 4. Polymers **C1**, **C2**, **C5**, and **C9** were too soft and/or brittle for testing machine, and it was impossible to detect the break during the measurement. The highest values of Young’s modulus and the lowest values of elongation and break were demonstrated by polymer **C4**, which was prepared with the lowest amount of C13-MA and BDT (60.05 MPa and 5.47%). Polymer **C3**, which was prepared with the same amount of VDM and C13MA as polymer **C4**, but with a higher amount of thiol, demonstrated a lower Young’s modulus value and greater elongation at break value (12.44 MPa and 17.3%). This behaviour was caused by the highly flexible thioether linkages. These linkages also explain why polymers with higher amounts of thiol stretched more during the tensile test and their elongation at break value was higher. Tridecyl methacrylate had a similar effect as thiol on the mechanical properties of the polymers. The increased amount of C13-MA in the composition led to the lower values of Young’s modulus. For example, by increasing the amount of C13-MA from 1.5 to 3 and then to 4.5 mol the Young’s modulus decreased as follows, from 60.05 MPa (polymer **C4**) to 23.033 MPa (polymer **C8**) and, finally, to 5.53 MPa (polymer **C12**). The elongation at break increased with increasing C13-MA in the resin from 5.47% (polymer **C4**) to 6.49% (polymer **C8**) and, finally, to 8.38% (polymer **C12**). This behaviour can be explained by the long carbon chain of tridecyl methacrylate. As a result, the polymer can stretch more, and is softer and more flexible [34]. 

### 3.7. Characterization of Microtransfered Structures

Two photocurable resins forming nonbrittle polymers, one rigid (**C8**) and the other soft (**C1**), have been tested in microtransfer molding (μTM). The results are depicted in Figure 8. Image (a) demonstrates the 3D printed USAF target. The thickest 3D printed lines were approximately 220–230 μm width, meanwhile the thinnest lines were about 68–69 μm width. Image (b) shows the PDMS stamp, which coincided well with the 3D printed target. The following images (c) and (d) depict replicas made of **C1** and **C8** resins. In both cases, the replicas corresponded to the PDMS mold, keeping its shape and all features (lines, letters, and numbers). Only minor drawbacks were observed on the replicas, which were voids (usually appearing due to air bubbles while dropcasting the resin) and loss of sharp edges due to peeling of the stamp. Despite this, both **C1** and **C8** resins showed great potential to be used for the μTM technique. 

## 4. Conclusions

Novel thermo-responsive shape-memory vanillin-based photopolymers have been developed and applied in microtransfer molding. Different mixtures of the two biobased monomers vanillin dimethacrylate and tridecylmethacrylate, to which 1,3-benzenedithiol has been added, have been tested as photocurable resins. The reduction of the thiol content was determined to increase the shrinkage of the polymers, the photocuring rate, and the values of the thermal and mechanical characteristics of the resulting polymers. The increase in tridecyl methacrylate content resulted in less rigid polymers, lower photocuring rate, and lower glass transition temperature. All polymers demonstrated great shape-memory properties and were able to return to their primary shape after the temperature programing. Furthermore, they were able to maintain their temporary shape when the temperature was lower than their glass transition temperature. Two photocurable resins forming nonbrittle polymers, one rigid and the other soft, have been tested in microtransfer molding and both resins demonstrated perfect replication, proving the novel bio-based photoresins to be suitable for microtransfer molding technique. 

The developed thermo-responsive shape-memory bio-based photopolymers have great potential for forming microtransfered structures and devices that can be used on nonflat surfaces. This work offers a sustainable route for exploiting biorenewable materials to replace established commercial photoresins applied in stamping technologies that are already being scaled up for cost-effective everyday use. 

## Figures and Tables

**Figure 1 polymers-14-02460-f001:**
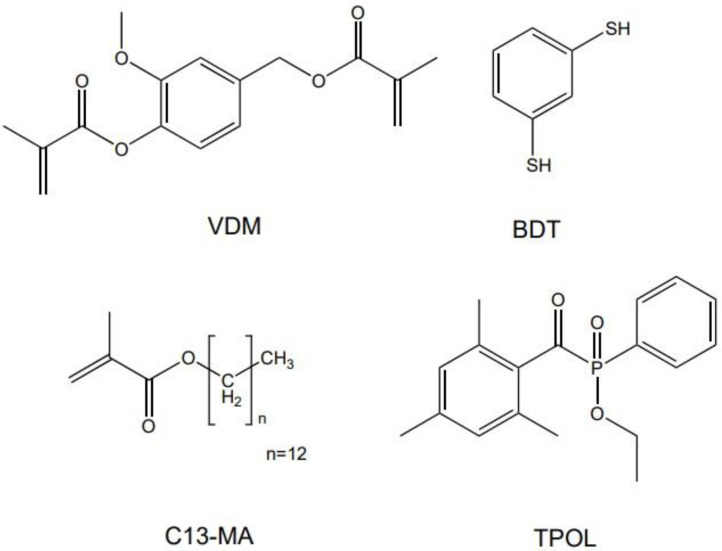
Chemical structures of vanillin dimethacrylate (VDM), 1,3-benzenedithiol (BDT), tridecyl methacrylate (C13-MA), and ethyl(2,4,6-trimethylbenzoyl)phenylphosphinate (TPOL).

**Figure 2 polymers-14-02460-f002:**
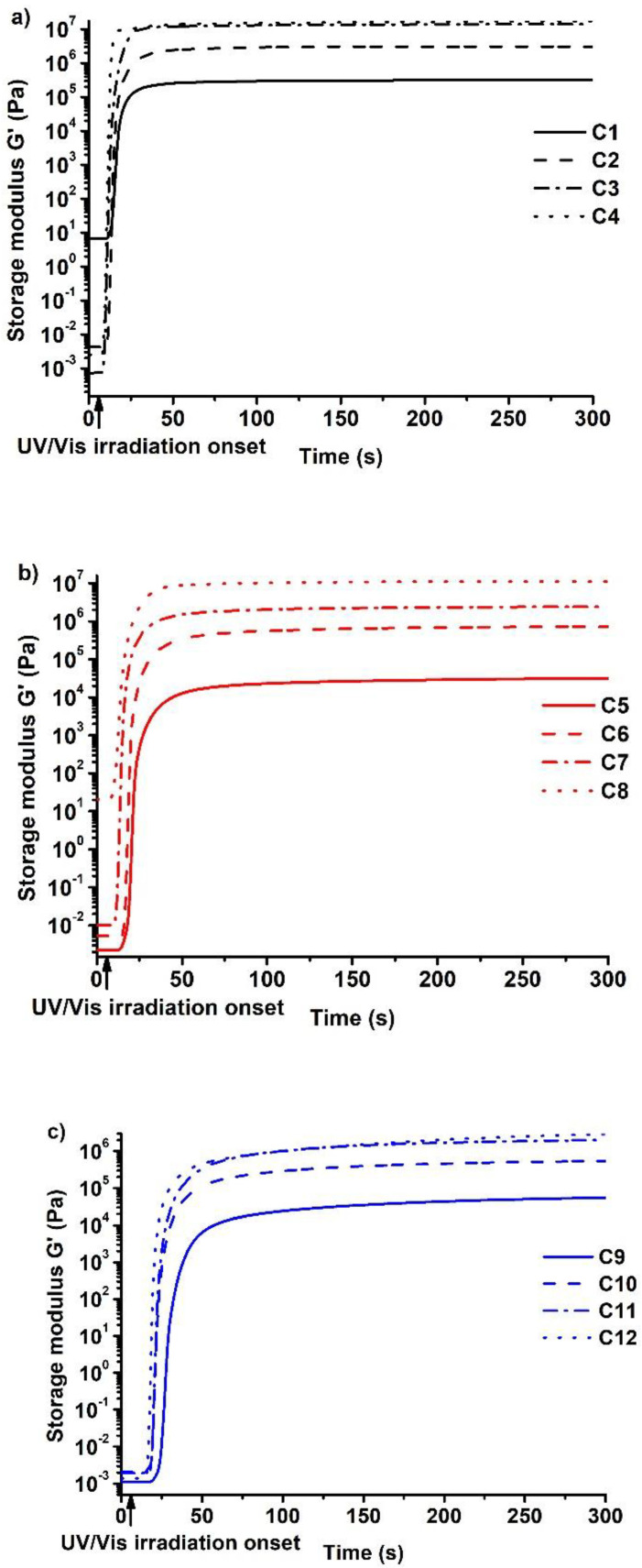
Dependency of storage modulus G′ of the resins **C1**–**C4** (**a**), **C5**–**C8** (**b**), and **C9**–**C12** (**c**) on the irradiation time.

**Figure 3 polymers-14-02460-f003:**
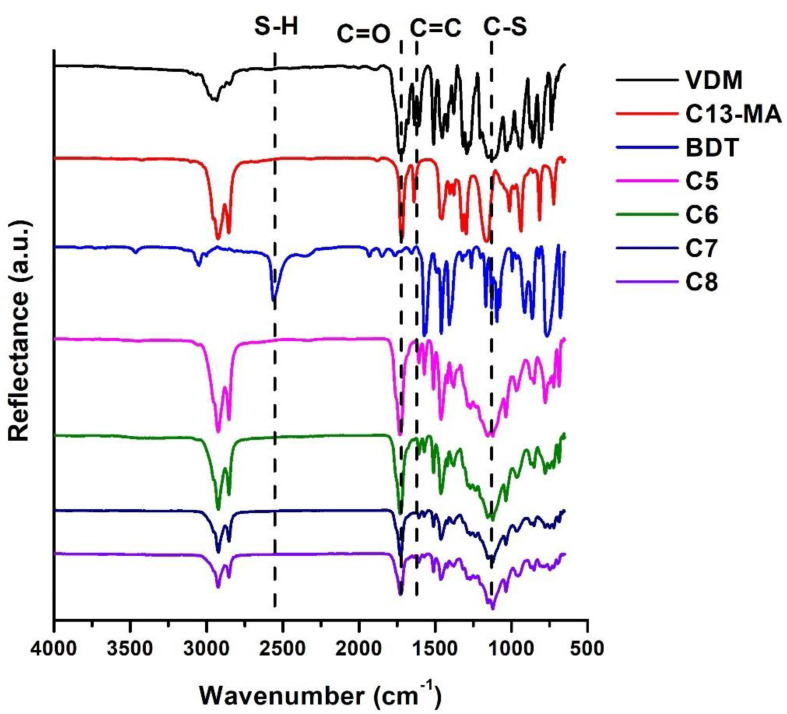
FT-IR spectra of VDM, C-13MA, BDT, and cross-linked polymers **C5**–**C8**.

**Figure 4 polymers-14-02460-f004:**
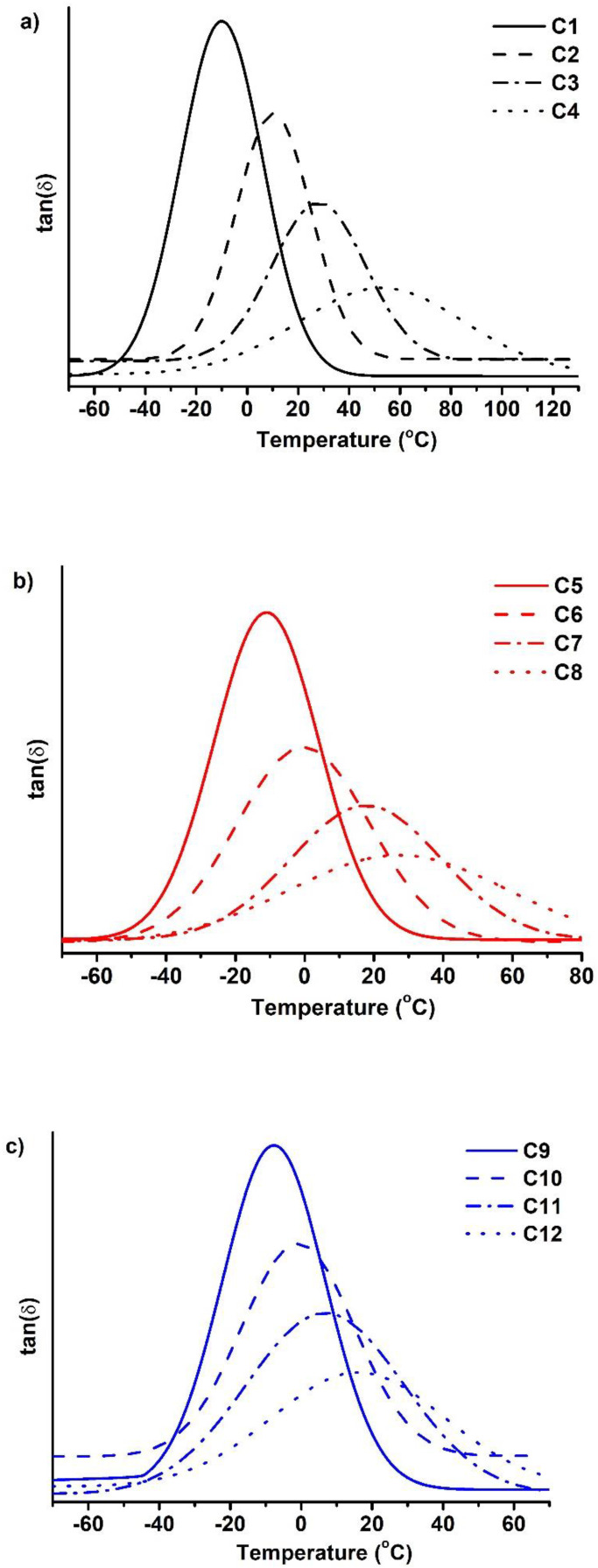
DMTA thermograms of cross-linked polymers **C1**–**C4** (**a**), **C5**–**C8** (**b**), and **C9**–**C12** (**c**).

**Figure 5 polymers-14-02460-f005:**
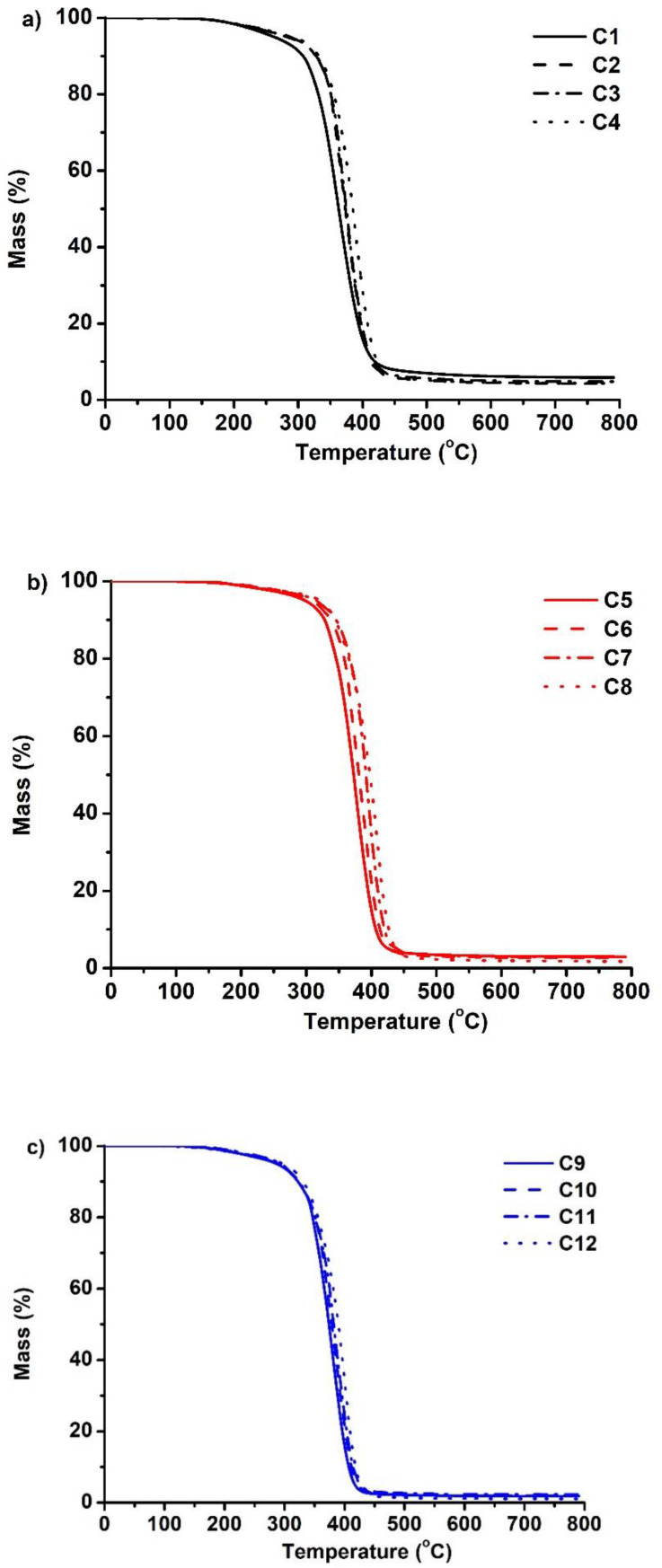
Thermogravimetric curves of cross-linked polymers **C1**–**C4** (**a**), **C5**–**C8** (**b**), and **C9**–**C12** (**c**).

**Figure 6 polymers-14-02460-f006:**
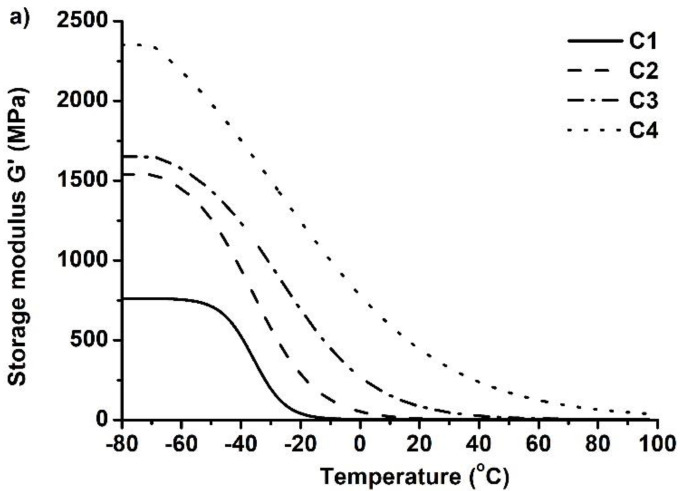
Dependency of the storage modulus G′ of polymers **C1**–**C4** (**a**), **C5**–**C8** (**b**), and **C9**–**C12** (**c**) on temperature.

**Figure 7 polymers-14-02460-f007:**
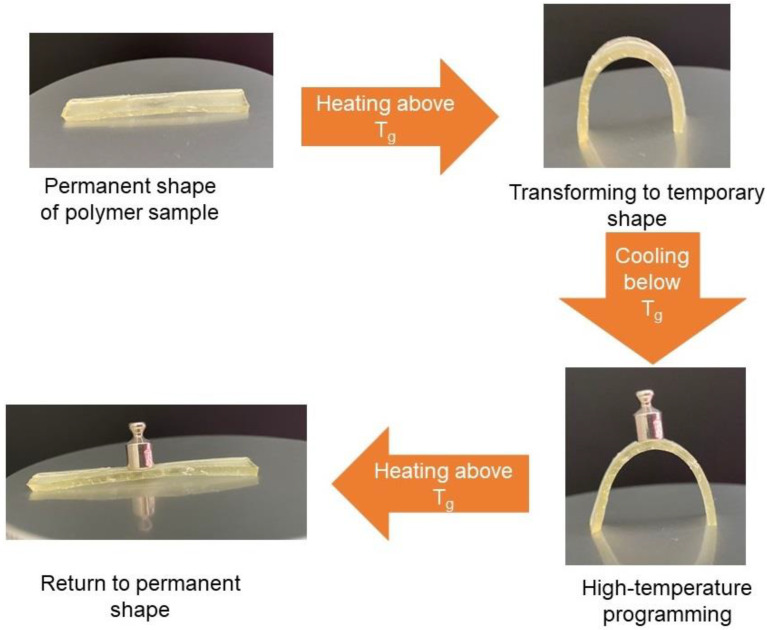
Scheme of shape-memory behaviour of polymer samples.

**Figure 8 polymers-14-02460-f008:**
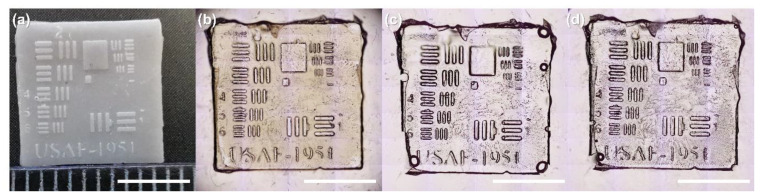
Results of μTM. (**a**) 3D printed 1951 USAF target (25% of the downloaded vector file size). (**b**) Soft PDMS mold. (**c**) and (**d**) are replicas made from **C1** and **C8** resins, respectively. Scale bars are the same in all the images and represent 5 mm.

**Table 1 polymers-14-02460-t001:** Composition of resins **C1**–**C12**.

Resin	Amount of VDM, Mol	Amount of BDT, Mol	Amount of C13-MA, Mol	Amount of TPOL, Mol.%
**C1**	1	1	1.5	3
**C2**	1	0.75	1.5	3
**C3**	1	0.5	1.5	3
**C4**	1	0.25	1.5	3
**C5**	1	1	3	3
**C6**	1	0.75	3	3
**C7**	1	0.5	3	3
**C8**	1	0.25	3	3
**C9**	1	1	4.5	3
**C10**	1	0.75	4.5	3
**C11**	1	0.5	4.5	3
**C12**	1	0.25	4.5	3

**Table 2 polymers-14-02460-t002:** Rheological Characteristics of Resins.

Resin	Storage Modulus, G′, MPa	Loss Modulus, G″, MPa	Loss Factor, tanδ	Complex Viscosity η*, MPa·s	Gel Point t_gel_, s	Induction Period, s	Shrinkage, %
**C1**	0.312	0.128	0.409	0.054	8.9	4.9	2
**C2**	3.030	2.061	0.680	0.635	6.8	2.6	7
**C3**	14.173	5.872	0.414	2.440	4.6	0.7	8
**C4**	17.343	5.661	0.327	2.900	3.4	0.3	9
**C5**	0.031	0.020	0.646	0.005	13.5	9.2	0
**C6**	0.721	0.333	0.461	0.013	10.6	7.3	2
**C7**	2.420	1.290	0.534	0.437	5.8	5.1	3
**C8**	10.148	4.470	0.442	1.760	3.7	0.9	4
**C9**	0.055	0.018	0.328	0.009	20.8	12.3	1
**C10**	0.542	0.230	0.425	0.094	14.4	11.9	3
**C11**	2.002	1.010	0.502	0.357	13.6	11.6	6
**C12**	2.930	1.450	0.497	0.520	9.8	8.8	10

**Table 3 polymers-14-02460-t003:** Yield of insoluble fraction, thermal, and thermomechanical characteristics of the polymers.

Resin	Yield of Insoluble Fraction, %	T_dec.−10%_, °C *	T_g_, °C **	Storage Modulus G′ at −80 °C, MPa ***	Storage Modulus G′ at 100 °C, MPa ****
**C1**	69	308	−10	761.99	3.36
**C2**	84	328	10	1539.61	1.17
**C3**	87	329	28	1649.52	3.90
**C4**	90	332	54	2350.37	21.67
**C5**	61	328	−9	386.44	0.19
**C6**	82	338	0	499.35	0.88
**C7**	88	346	18	778.79	2.22
**C8**	85	344	28	902.59	4.84
**C9**	70	322	−7	19.08	0.58
**C10**	81	323	−1	911.49	3.15
**C11**	88	324	7	890.68	2.27
**C12**	79	331	17	661.37	2.69

* from TGA curves. ** from DMTA curves. *** before glass transition temperature. **** after glass transition temperature.

**Table 4 polymers-14-02460-t004:** Mechanical characteristics of the polymers obtained by the tensile test.

Resin	Young’s Modulus, MPa	Tensile Strength, MPa	Elongation at Break, %
**C1**	- *	-	-
**C2**	-	-	-
**C3**	12.436	1.256	17.30
**C4**	60.050	1.976	5.47
**C5**	-	-	-
**C6**	1.865	0.140	12.19
**C7**	3.991	0.309	11.10
**C8**	23.033	0.960	6.49
**C9**	-	-	-
**C10**	2.317	0.040	22.36
**C11**	3.054	0.129	10.71
**C12**	5.531	0.227	8.38

*—not determined due to excessively soft and/or brittle polymer.

## Data Availability

Not applicable.

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
