# Peer review of "Thermo-Responsive Shape Memory Vanillin-Based Photopolymers for Microtransfer Molding"

_polymers, 2022, doi:10.3390/polym14122460_

Round 1
Reviewer 1 Report
In this manuscript,authors developed thermo-responsive shape memory vanillin-based photopolymers and applied to these materials to micro-transfer molding. Although some results have been provided, I can not recommend to accept it for publication in the present format. The detailed comments are shown below:
(1) Figures 2 -7 are vague, should be redrawn.
(2) The full name and the acronym are confusing, In any case give the full name first, and then use acronym in the following parts.
(3) A lower amount of thiol and a higher amount of acrylate lead to seriously shrink, why? More discussion should be provided.
(4) Authors think the spatial hindrances lead to a relatively low value of the yield of the insoluble fraction. The explanation needs to be further verified by characterizing the composition of soluble fraction.
Author Response
The response to the Reviewer 1 comments is attached.

Reviewer 2 Report
Paper "Thermo-responsive shape memory vanillin-based photopolymers for micro-transfer molding" presents interesting results describing the thermo-responsive shape memory vanillin-based photopolymers for microtransfer molding. Paper can be accepted for publication in Polymer mdpi after major revision.
First of all, the authors should point out what was new compared to previous papers "Vanillin acrylate-based thermo-responsive shape memory antimicrobial photopolymer" and "Influence of vanillin acrylate-based resin composition on resin photocuring kinetics and antimicrobial properties of the resulting polymers". I believe the presented paper includes principal new results but this information should be emphasized strictly.
It will be valuable to add a scheme to illustrate the processes of synthesis of the thermo-responsive shape memory vanillin-based photopolymers in detail.
Please add an appropriate discussion in the Introduction about thermo-responsive bio-based polymer systems which are based on glass transition temperature. I recommend to cite suitable papers:
https://doi.org/10.1021/acs.langmuir.6b02946
https://doi.org/10.1016/j.apsusc.2017.03.001
https://doi.org/10.1021/acs.macromol.7b01889
The sentence "In this work, different mixtures of bio-based monomer vanillin dimethacrylate with tridecyl methacrylate, derived from natural oil....." should be explained.
Author Response
The response to the Reviewer 2 comments is attached.

Round 2
Reviewer 2 Report
After revision, the paper can be accepted in present form.